# Proteome of Glioblastoma-Derived Exosomes as a Source of Biomarkers

**DOI:** 10.3390/biomedicines8070216

**Published:** 2020-07-16

**Authors:** Stanislav Naryzhny, Andrey Volnitskiy, Arthur Kopylov, Elena Zorina, Roman Kamyshinsky, Viktor Bairamukov, Luiza Garaeva, Anatoly Shlikht, Tatiana Shtam

**Affiliations:** 1Orekhovich Institute of Biomedical Chemistry of Russian Academy of Medical Sciences Pogodinskaya 10, 119121 Moscow, Russia; a.t.kopylov@gmail.com (A.K.); el.petrenko@bk.ru (E.Z.); 2Petersburg Nuclear Physics Institute NRC «Kurchatov Institute», Orlova Roshcha 1, 188300 Gatchina, Russia; voln.a@yandex.ru (A.V.); vbayramukov@gmail.com (V.B.); garaeva.luiz@yandex.ru (L.G.); 3National Research Center “Kurchatov Institute”, Akademika Kurchatova pl. 1, 123182 Moscow, Russia; kamyshinsky.roman@gmail.com; 4Shubnikov Institute of Crystallography of Federal Scientific Research Centre ’Crystallography and Photonics” of Russian Academy of Sciences, Leninskiy Prospect 59, 119333 Moscow, Russia; 5Moscow Institute of Physics and Technology, Institutsky Lane 9, Dolgoprudny, 141700 Moscow, Russia; 6Peter the Great Saint-Petersburg Polytechnic University, Politehnicheskaya 29, 19525 St. Petersburg, Russia; 7Far Eastern Federal University, Sukhanova 8, 690091 Vladivostok, Russia; schliht@mail.ru

**Keywords:** glioblastoma, protein expression, proteomics, biomarkers, exosomes

## Abstract

Extracellular vesicles (EV) are involved in important processes of glioblastoma multiforme (GBM), including malignancy and invasion. EV secreted by glioblastoma cells may cross the hematoencephalic barrier and carry molecular cargo derived from the tumor into the peripheral circulation. Therefore, the determination of the molecular composition of exosomes released by glioblastoma cells seems to be a promising approach for the development of non-invasive methods of the detection of the specific exosomal protein markers in the peripheral blood. The present study aimed to determine the common exosomal proteins presented in preparations from different cell lines and search potential glioblastoma biomarkers in exosomes. We have performed proteomics analysis of exosomes obtained from the conditioned culture medium of five glioblastoma cell lines. A list of 133 proteins common for all these samples was generated. Based on the data obtained, virtual two-dimensional electrophoresis (2DE) maps of proteins presented in exosomes of glioblastoma cells were constructed and the gene ontology (GO) analysis of exosome proteins was performed. A correlation between overexpressed in glial cell proteins and their presence in exosomes have been found. Thus, the existence of many potential glioblastoma biomarkers in exosomes was confirmed.

## 1. Introduction

The glioblastoma multiforme (GBM) is the most malignant form of brain tumors and one of the most aggressive types of cancer [1]. As the median survival time for patients with glioblastoma ranges from 12 to 15 months, and the 2-year overall survival rate averages only 26%, early diagnosis, and treatment is important to improve outcome [2]. Despite the efforts of the medical and scientific community, a reliable non-invasive method for early diagnosis of glioblastoma has not been developed up to date. Computed tomography and brain biopsy are still the main methods for the diagnosis of this disease. Especially, the need for biomarkers is becoming more apparent as the clinical management of GBM moves towards personalized medicine. There is a hope that EV secreted by glioblastoma cells could be a source of biomarkers detectable in the peripheral circulation [3]. EV are the particles released either from the cell surface–microvesicles having a size of 100–1000 nm or from an endosomal route–exosomes having a size of 40–150 nm [4]. EV contain an array of different types of macromolecules–proteins, nucleic acids, lipids, and metabolites–which could be considered as the candidates for biomarkers [5]. Available information indicates that more invasive GBM cells secrete higher amounts of exosomes. This strategy allows tumors to hijack their microenvironment and modulate anti-tumor immunity [3,6]. As EV can cross the hematoencephalic barrier and enter the peripheral circulation, macromolecular profiling of GBM-derived EV could be a noninvasive way of assessing tumors in situ. Extensive studies of RNA content in glioma-derived EV were performed [7,8,9]. In several studies, an array of protein cargo of GBM-derived EV has been identified, however, a slender picture of the EV protein profiles is not yet available [10,11,12]. Previously, we have performed a comparative analysis of proteins from glioblastoma and normal cells trying to select the potential cancer biomarkers. Based on this analysis, a list of proteins overexpressed in glioblastoma cells as “potential glioblastoma biomarkers” was generated [13]. We have constructed a database of proteoforms for glioblastoma as well as for normal fibroblasts [14]. Here, we provide a graphical protein pattern (virtual 2DE) of exosomes secreted from five glioblastoma cell lines using high-resolution mass spectrometry (MS) and discuss the presence and the functional roles of the proteins from glioblastoma cells in exosomes. Additionally, we show that many proteins with enhanced expression levels in glioblastoma cells are presented in the exosome pattern. The gene ontology (GO) analysis of the exosome protein pattern was also performed.

## 2. Results

### 2.1. Characterization of Extracellular Vesicles Isolated from Conditioned Culture Medium of Glioma Cell Lines

Five samples of vesicles isolated from the conditioned culture medium (CCM) of glioma cell lines were analyzed by Dynamic Light Scattering (DLS), Nanoparticle Tracking Analysis (NTA), and Atomic Force Microscopy (AFM). The presence of exosome surface markers CD9 and CD81 in the membrane of the vesicles was evaluated by flow cytometry. Finally, a pooled sample of the isolated vesicles was examined using Cryo-Electron Microscopy (cryo-EM). DLS was used to estimate the average hydrodynamic radius of the isolated vesicles. An example of determining the particle size distribution present in a sample of EV from CCM of Glia-Tr cells is shown in Figure 1A. Thus, a major peak corresponding to an exosome size of about 100 nm is presented in all the vesicle samples isolated from CCM of glioma cells. In addition, various sizes of vesicles such as a peak of about 10–20 nm corresponding to contaminating proteins, as well as a peak of large particles with sizes of 2000–5000 nm were observed in all the five analyzed samples of EV from glioma cells. Next, NTA was applied to confirm the measurement of EV size and to estimate their concentration (Figure 1B). The mode size of the vesicles isolated from CCM of glioma cells was in a range from 60 to 85 nm, which corresponded well with the size of exosomes [4]. The concentration of vesicles in the final ultracentrifuged samples was approximately the same for all studied samples and amounted to 2–4 × 10^13^ particles/mL. To confirm the exosome nature of isolated particles, they were non-specifically absorbed on the flow cytometry beads followed by incubation with Alexa488-conjugated or APC-conjugated antibodies to tetraspanins CD9 or CD81. The presence of exosome markers on the surface of the vesicles was estimated in parallel in five samples of vesicles isolated from CCM of glioma cells using the same exosome standard as a positive control and beads non-incubated with any vesicles as a negative control (Figure 1C). The presence of CD9 or CD81-positive vesicles was confirmed by flow cytometry in all the samples. Later, both exosome markers were identified by mass spectrometric analysis in all the samples of vesicles isolated from 5 glioma cell lines (Appendix A).

The morphology of EV was studied by AFM. All the samples demonstrated the abundance of individual particles of a near-spherical shape with the size ranging from 40 to 140 nm (Figure 2). The enlarged area illustrates the morphology of a single exosome (1) and its substructural features obtained in a phase shift mode (2). A cup-shape effect appeared due to the uneven response of the central area of exosome and its surroundings to the mechanical perturbation exerted by the AFM tip.

To confirm the vesicular nature of particles isolated from CCM the pooled samples were analyzed by cryo-EM. Figure 3A,B demonstrates that most isolated particles have a round-shaped vesicular morphology formed by a characteristic lipid bilayer with an average thickness of ∼5–8 nm (Figure 3C). The visualized particles generally varied in size from 20 to 350 nm with an average of 103 ± 62 nm (Figure 3H). Several types of vesicles were observed: single (Figure 3C), double (Figure 3F), and multilayer vesicles (Figure 3D). Single and double vesicles containing electron-dense material were also visualized (Figure 3E,G). Similar types of EV were described earlier in the ejaculate, blood plasma, and cerebrospinal fluid [15,16,17]. A small part of the observed particles was round, about 20 nm in size and without a visible membrane structure, which indicates some contamination of the samples with protein aggregates or lipoproteins. In addition, the presence of large vesicles (more than 200 nm) was also observed (Figure 3A,B,D,F,H).

### 2.2. Proteome Profiling of Glioblastoma-Derived Extracellular Vesicles

Overall, 896 different proteins/isoforms coded by 880 genes (Appendix A) were identified in exosome preparations from five different glioma cell lines (from Glia-Sh–373, Glia-L–482, A172–536, Glia-R–385, Glia-Tr–367). Among them, 133 proteins were common to exosomes secreted by all the cell lines (Figure 4).

To investigate the functional basis of the identified exosomal proteins from CCM of glioma cell lines, we have analyzed and classified their molecular functions. The annotated biological functions of the proteins revealed enrichment of glioma exosome-associated proteins related to “metabolism,” “cell adhesion”, “protein biosynthesis,” “chaperones”, “G-proteins and GTPase” and “cytoskeleton” (Figure 5).

To obtain a better visual representation of the protein pattern a virtual 2DE was applied. Such a representation allows to compare the patterns of exosomal (Figure 6) and cellular proteins (Figure 7) to figure out whether the pattern of exosome proteins could be mainly the reflection of the pattern of the most abundant cellular proteins [13]. But, as we can see from Figure 6 and Figure 7, these patterns look different. Even though many exosome proteins are the major cellular proteins (actin, vimentin, tubulin, G3P…), there are many non-abundant proteins. It indicates that proteins are selected for cargo not randomly but rather by a specific mechanism.

Previously, we analyzed the proteins and their proteoforms in glioblastoma cells [13,18,19]. A list of “potential glioblastoma biomarkers” was generated in that study [13]. It happened that 11 proteins from this list [13] are present in all the five exosome samples analyzed in the present study (Appendix A). These proteins are pyruvate kinase PKM (KPYM), annexin A1 (ANXA1), triosephosphate isomerase (TPIS), vimentin (VIME), annexin A2 (ANXA2), transitional endoplasmic reticulum ATPase (TERA), alpha-enolase (ENOA), peroxiredoxin-1 (PRDX1), glyceraldehyde-3-phosphate dehydrogenase (G3P), heat shock protein HSP 90-beta (HS90B), 14-3-3 protein epsilon (1433E). Two more proteins, nucleophosmin (NPM) and cofilin (COF1), are present in four exosome samples (Appendix A).

The identified exosomal proteins were analyzed by PANTHER, a software that allows us to classify proteins and their genes. The detected proteins were classified according to their involvement in a biological process (Figure 8). Interestingly, the graphs for all the detected (896) exosome proteins (Figure 8A) and the common exosome (133) proteins (Figure 8B) are almost the same. What is more, even the graph for only 13 potential biomarkers looks very similar (Figure 8C). The most notable biological processes are “cellular process” (GO:0009987), “biological regulation” (GO:0065007), “metabolic process” (GO:0008152), and “cellular component organization or biogenesis” (GO:0071840). A more detailed analysis of the involvement of these proteins in the cellular processes is presented in Appendix A. Similar results for protein functions in glioblastoma exosomes have been described in the literature [20,21]

## 3. Discussion

Exosomes promote tumor formation and development by mediating the intercellular transport of miRNAs, mRNA, and proteins, and there is growing interest in their use as biomarkers for disease diagnosis and the monitoring of disease recurrence [22]. Exosomes secreted by cancer cells stimulate the malignancy of gliomas, including suppressing the immune response or affecting the tumor microenvironment [23,24,25]. Exosomes released by glioblastoma cells may cross the hematoencephalic barrier and carry molecular cargo into the peripheral circulation. Therefore, the determination of the molecular composition of exosomes secreted by glioblastoma cells seems to be an extremely promising direction for the development of non-invasive methods for diagnosing this disease [26]. In this study, we performed a proteomic analysis of vesicles isolated from the conditioned medium of five glioma cell lines. EV from glioma cells have been isolated by ultracentrifugation and characterized in size, quantity, and morphology by commonly used in nanoparticle size and shape interpretation methods, including NTA, DLS, atomic force, and cryo-EM microscopy. According to the International Society for Extracellular Vesicles (ISEV) guidelines, isolated EV are considered to be exosomes if they are in the size range of 30–200 nm, have a typical spherical form, contain a bilayer membrane, and are enriched with exosomal markers [27]. The results obtained from DLS or NTA demonstrate the presence of particles with a characteristic exosome size in the final samples [4]. Cytometric and MS detection of the CD9 or CD81 markers on the surface of the particles confirms the exosomal nature of the vesicles. Moreover, most particles detected by both cryo-EM and AFM have a vesicular structure and match the dimensional characteristics of exosomes. Summarizing the data, it can be stated that most vesicles isolated from CCM of glioma cells can be defined as exosomes.

A set of 133 proteins that are common in all the five samples of exosomes secreted by GBM cell lines was detected. Correlating each protein with its main function, whole groups of proteins were identified that were associated both with processes occurring in tumor cells and with possible functions of exosomes secreted by them. First, we drew attention to proteins that may be related to the formation or traffic of exosomes. These included structural vesicular proteins, for example, clathrin (CLTC), chaperones involved in the assembly of vesicles, cytoskeleton components, small G-proteins and other proteins involved in the movement of vesicles and their fusion with the membrane. Some of them are of particular interest—for example, vimentin (VIME) involved in the positioning of organelles is a marker of the epithelial-mesenchymal transition of tumor cells [28]. Programmed cell death 6-interacting protein, PDCD6IP, interacts with endophilins that regulate the shape of the membrane during endocytosis and protects cells from Ca2 + -dependent apoptosis [29]. It was shown that increased expression of this gene and endophilins leads to vacuolization of cells, protecting them from programmed death [30,31].

It was surprising that the whole set of the nucleic acid interacting proteins, including several chromatin and ribosomal proteins, fell into the set of the proteins common to exosomes from glioma cell lines. As part of exosomes, these proteins can be absorbed by macrophages. Presented to lymphocytes, fragments of these proteins can participate in the formation of immunological tolerance to tumor cells. Among the proteins that regulate the behavior of vesicles, we identified a vasolin-containing protein, TERA(VCP), ATPase, which is needed to remove improperly assembled or spent peptides from ribosomes, the endoplasmic reticulum, mitochondria, and chromatin. TERA(VCP)is involved in membrane fusion and the regulation of vesicle traffic from the plasma membrane to the lysosome. We assume that a sorting mechanism is associated with the TERA(VCP) function, which explains the presence of the above proteins in exosomes.

Among the proteins identified in the five exosome samples, the cell adhesion proteins include integrins, annexins, tetraspanins (including exosome markers CD9 and CD81), proteases, and protease inhibitors. These proteins can affect the microenvironment of tumor cells, promoting processes such as invasion, angiogenesis, and the modulation of the immune response (e.g., vitronectin, VTN, which inhibits the complement system). Among these proteins, ANXA2, CD44, and tenascin-C (TENA) were identified. Annexin A2 (ANXA2) plays a significant role in the invasion, metastasis, angiogenesis, proliferation, actin polymerization, and endosomal sorting [32]. CD44 is a receptor for hyaluronic acid and can also interact with other ligands, such as osteopontin, collagens, and matrix metalloproteinases. It is implicated in various malignant processes including cell motility, tumor growth, and angiogenesis, and has been suggested as a cancer stem cell marker [33]. Tenascin-C (TENA) is a hexameric glycoprotein expressed by the neural stem and progenitor cells and endothelium during embryogenesis. It is located in the extracellular matrix and influences cell migration, inhibits focal contact formation, and promotes angiogenesis. TNC is overexpressed and correlated with tumor malignancy observed in many cancer types and is associated with glioma outcomes [34]. Previously, we have shown the increased levels of stem cell genes, including *SOX2*, *OCT4*, and *GLI1* in these glioma cell lines [35,36] that coincide with the presence of the CD44 and TNC in all the exosome samples from CCM.

Proteins involved in metabolic processes are mainly represented by groups of enzymes, ion channels, amino acid transporters, and subunits of G-proteins that regulate adenylate cyclase and phospholipase C. The set of enzymes contained in exosomes of glioma cells well reflects the characteristics of the metabolism of tumor cells, most of which catalyze various stages of glycolysis and are involved in the Warburg effect. So, among the proteins found in all the exosome samples, we identified lactate dehydrogenase B (LDHB), which converts pyruvate to lactic acid, and ATP, as well as phosphoglycerate kinase 1 (PGK1), which not only is responsible for the synthesis of 3-phosphoglycerate, but also mediates the inhibition of citric acid synthesis [37]. It was shown that glioma cell-derived exosomes activate glycolysis in human bone marrow mesenchymal stem cells, resulting in their tumor-like phenotype transformation [25]. Interestingly, the glioma exosomes contained ATP-citrate synthase (ACLY), which converts citric acid into acetyl coenzyme A, and fatty acid synthase (FASN), which synthesizes palmitate from acetyl coenzyme A in the presence of NADPH. This indicates the possibility of a transition between different branches of energy metabolism, which probably creates an additional opportunity for tumor cells to adapt to adverse conditions.

Additionally, we draw attention to two proteins that can participate in the development of glioblastomas and their resistance to treatment. Firstly, it is the DNA-dependent protein kinase catalytic subunit (PRKDC), which is involved in the repair of double-strand breaks with a nonhomologous DNA end joining (NHEJ) contributing to radioresistance [38]. It has been shown that the degradation of this protein is regulated by the vasolin-containing protein, VCP, another protein found in the exosome samples [39]. Secondly, it is a major vault protein (MVP), which is associated with nuclear pores, regulates nuclear-cytoplasmic transport and determines resistance to certain drugs [40,41]. It should be noted that its gene is a target of the transcription factor GLI1 [42], which according to our previous data is abnormally active in these glioma lines [36].

An important conclusion comes after a comparative analysis of exosomal proteins and “potential glioblastoma biomarkers”. Based on our previous analysis of the proteomes of the glioma and normal cells, a list of proteins overexpressed in glioma cell lines was generated [13,18]. It is important that 13 of them (ANXA1, ANXA2, ENOA, G3P, HS90B, KPYM, PRDX1, TPIS, TERA, VIME, 1433E, COF1, NPM) were detected by MS in glioblastoma exosome samples (Figure 5 and Appendix A). We can call these proteins “exosomal glioblastoma biomarkers”. The data obtained in the present study allows us to expand the list of exosomal potential biomarkers of gliomagenesis by adding CD44 and Tenascin-C (TNC). Both these proteins are among the 133 common exosomal proteins identified in our study and are associated with malignancy of gliomas [33,34]. Surprisingly, such proteins as PCNA and p53, which are at the top of the list of “potential glioblastoma biomarkers” [13,18], were not detected by MS in exosome preparations. These proteins had been previously detected in glioblastoma exosomes by Western blotting [19]. These proteins were detected in exosomes from ovarian cancer cells and are presented in Vesiclepedia http://microvesicles.org/index.html#. We can conclude that our list of exosomal proteins of glioblastoma detected by MS is not complete, and immunological methods of detection can sometimes be more sensitive than MS.

In summary, we can conclude that the set of 133 proteins common to glioblastoma exosomes detected in our study is only partial but represents the main functional aspects of exosome cargo proteins and could be representative for exosomes functions. Our study suggests the use of specific brain tumor exosomes and several protein markers to help to create non-invasive techniques to diagnose disease. According to our data, exosomes contain reliable protein biomarkers of glioblastoma.

## 4. Materials and Methods

### 4.1. Glioma Cell Lines and Cultivation Conditions

The work was carried out using five glioma cell lines: A172 cell line obtained from the collection of the Institute of Cytology of the Russian Academy of Sciences (St. Petersburg, Russia) and glioma cell lines (Glia-Tr, Glia-L, Glia-R, and Glia-Sh) generated in the Laboratory of Cell Biology (NRC «Kurchatov Institute»-PNPI, Gatchina, Russia) [35,36]. The cells were cultured in an atmosphere of 5% CO_2_ at 37 °C in DMEM/F12 (1:1) medium containing L-glutamine (BioloT, Saint Petersburg, Russia) and supplemented with 5% fetal bovine serum purified from exosomes by ultracentrifugation at 110,000× *g* for 18 h. The conditioned culture medium (CCM) was harvested after 72–96 h of incubation and growth of a cell monolayer in it. Aliquots of the collected CM were stored at 4 °C in glass flasks, polled to a volume of 250 mL.

### 4.2. Isolation of Exosomes from the Conditioned Culture Medium (CCM)

EV were isolated from CCM by ultracentrifugation according to standard procedure as it was done in previous studies [12,43,44]. Exosomes were isolated from equal volumes (250 mL) of CCM as described before [12]. Briefly, after preliminary removal of cellular detritus and large vesicles by centrifugation (2000× *g* for 30 min, and then 16,000× *g* for 30 min) sequential ultracentrifugation at 110,000× *g* for 2 h (45Ti rotor) was performed. After centrifugation, the supernatant was removed and the pellet re-suspended in 1 mL of phosphate-buffered saline (PBS) for at least 1 h at 4 °C, the volume was adjusted to 5 mL and re-centrifuged at 110,000× *g* for 2 h (SW 55Ti rotor). The resulting pellets were re-suspended with gentle shaking in 100 mL of PBS for at least 1 h at 4 °C. Final samples of nanovesicles were aliquoted, rapidly frozen in liquid nitrogen, and stored at −80 °C until analysis.

### 4.3. Exosome Analysis

Isolated vesicles were characterized according to the International Society for Extracellular Vesicles (ISEV) guidelines [27]. The EV size distribution was evaluated by the method of dynamic light scattering (DLS) using the Zetasizer Nano ZSP system (Malvern Instruments, Malvern, UK). Measurements were carried out at +25 °C. For each sample, the particle size distribution curves were plotted by the results of three measurements. The analysis and visualization of the results were carried out using the Zetasizer Software.

The EV size and concentration were determined by nanoparticle tracking analysis (NTA) using the NTA NanoSight^®^ LM10 (Malvern Instruments) analyzer equipped with a blue laser (45 mW at 488 nm) and a C11440-5B camera (Hamamatsu Photonics K.K., Hamamatsu, Japan). To optimize the measurement mode, the samples of isolated vesicles were diluted 1:100, 1:1000, or 1:10,000 by PBS. In the selected dilution, each sample was measured in triplicate. Recording and data analysis were performed using the NTA software 2.3. The following parameters were evaluated during the analysis of recording monitored for 60 s: the average hydrodynamic diameter, the mode of distribution, the standard deviation, and the concentration of vesicles in the suspension.

Detection of EV was carried out by AFM. Briefly, the suspension of EV in PBS was diluted 50 times with Milli-Q, and 10 µL were dripped on the freshly cleaved mica. After 5 min of incubation, the mica was washed in Milli-Q and left for natural evaporation for 4 hrs. AFM measurements on the air were carried out using the Solver BIO microscope manufactured by NT-MDT (Russia), the frequency was 1 Hz. NSG01_DLC probes in tapping mode were used. Image processing was carried out using Gwyddion 2.49 software [45].

Cryo-electron microscopy (cryo-EM) was used for the direct visualization of vesicles as described previously [15]. The study was carried out on a Titan Krios 60–300 TEM/STEM (FEI, Hillsboro, OR, USA) transmission electron microscope equipped with direct electron detector Falcon II (FEI, USA) and a Cs image corrector (CEOS, Heidelberg, Germany).

Quantitative analysis of the exosome markers (tetraspanins CD9 or CD81) on the surface of the isolated EV was carried out using an Exo-FACS ready-to-use kit for the analysis of exosome markers from cell culture media (Lonza, Tallinn, Estonia) supplied with primary antibodies against CD9 and secondary Alexa488-labeled antibodies, according to the manufacturer’s recommendations. Additionally, bead-coupled EV were assayed using an APC-conjugated Anti-CD81 antibody (Beckman Coulter, Brea, CA, USA). The same number of vesicles was added to each sample for flow cytometry, based on the results of measurements of particle concentration using NTA. A sample without any EV was used as a negative control for non-specific labeling. An aliquot (5 mL, ≈10^13^ particles) of exosome standard was used as a positive control. The analysis was performed with a CytoFlex instrument (Beckman Coulter, USA).

### 4.4. Mass Spectrometry (MS)

MS (shotgun analysis) of exosomal proteins was performed according to the filter-aided sample preparation (FASP) protocol using the equipment of “Human Proteome” Core Facilities of the Institute of Biomedical Chemistry (Russia) [46]. In short, the proteins were digested with trypsin (Promega Trypsin Gold), tryptic peptides were dried in a vacuum centrifuge and dissolved in 5% (*v*/*v*) formic acid. Tandem mass spectrometry analysis of resulting peptides from each set was performed in duplicate using an Orbitrap Fusion Lumos MS (Thermo Scientific Waltham, MA, USA) [46,47,48]. Liquid chromatography-mass spectrometry (ESI LC-MS/MS) of the peptides was carried out using an Agilent HPLC Series 1100 chromatography system (Agilent Technologies, Santa Clara, CA, United States). Approximately 4 µg of peptides were added to a Zorbax 300SB-C18 5 × 0.3 µm trap column (Agilent Technologies). After washing with a 5% ACN solution containing 0.1% formic acid, the peptides were separated on a 150 mm × 75 µm back phase analytical Zorbax 300SB-C18 (Agilent Technologies) column by a 30-min gradient (5–60% ACN, 0.1% formic acid) at a flow rate of 300 nL/min. Mass spectra were collected in a positive ion mode. High-resolution data were obtained on the Orbitrap analyzer with a resolution of 30,000 (*m*/*z* 400) for MS scans and 7500 (*m*/*z* 400) for MS/MS scans. The data were analyzed by SearchGui v3.3.20 [49] using the following parameters: enzyme–trypsin; maximum of missed cleavage sites–2; fixed modifications–carbamidomethylation of cysteine; variable modifications–oxidation of methionine, phosphorylation of serine, threonine, tryptophan, acetylation of lysine; the range of the precursor mass error–20 ppm; the product mass error–0.01 Da. As a protein sequence database, NeXtProt (October 2014) was used. Only 100% confident results of protein identification with the detection of at least two peptides were selected for presentation.

### 4.5. Gene Ontology Functional Annotation and Pathway Analysis

Gene ontology analysis of exosomal proteins was performed. The groups of proteins that were identified were analyzed by program ShinyGO v0.61 using the Database for Annotation, Visualization and Integrated Discovery (DAVID 6.8) [50,51] and by PANTHER (Protein Analysis THrough Evolutionary Relationships; http://www.pantherdb.org). The proteins were classified according to their biological process and molecular function. Human molecular pathways were extracted from the following databases: BioCarta (https://cgap.nci.nih.gov/Pathways/BioCarta_Pathways), KEGG [52], NCI [53], Reactome [54], and SABiosciences Pathway Central (http://www.sabiosciences.com/pathwaycentral.php).

## Figures and Tables

**Figure 1 biomedicines-08-00216-f001:**
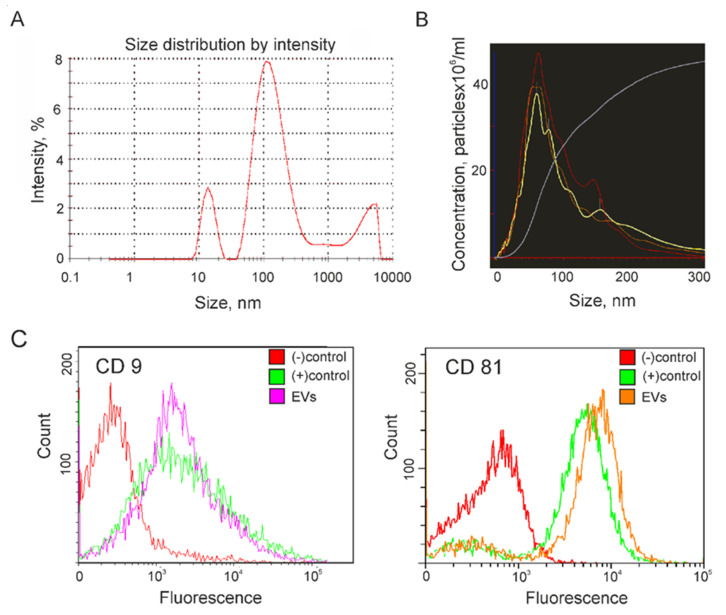
Characterization of extracellular vesicles (EV) isolated from the conditioned culture medium (CCM) of Glia-Tr cell line. (**A**) The size of EV was quantified by dynamic light scattering (DLS). (**B**) Nanotracking particle analysis (NTA) of particle size and concentration. Each sample was measured in triplicate. (**C**) Flow cytometry analysis of isolated EVs for the surface expression of exosome markers CD9 (left panel) and CD81 (right panel). Immunobeads that were not incubated with EV during sample preparation were used as a negative control ((−) control). An aliquot of exosome standard (Lonza) was used as a positive control ((+) control).

**Figure 2 biomedicines-08-00216-f002:**
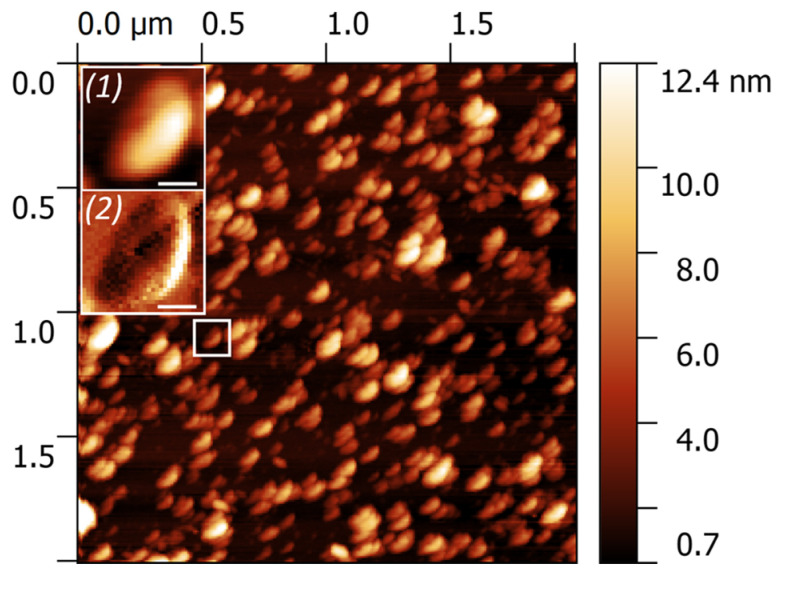
Large-scale atomic force microscopy (AFM) image of EV derived from CCM of glioma cells. The insertions demonstrate the height (**1**) and phase shift (**2**), revealing a cup-shape effect. The scale bar is 40 nm.

**Figure 3 biomedicines-08-00216-f003:**
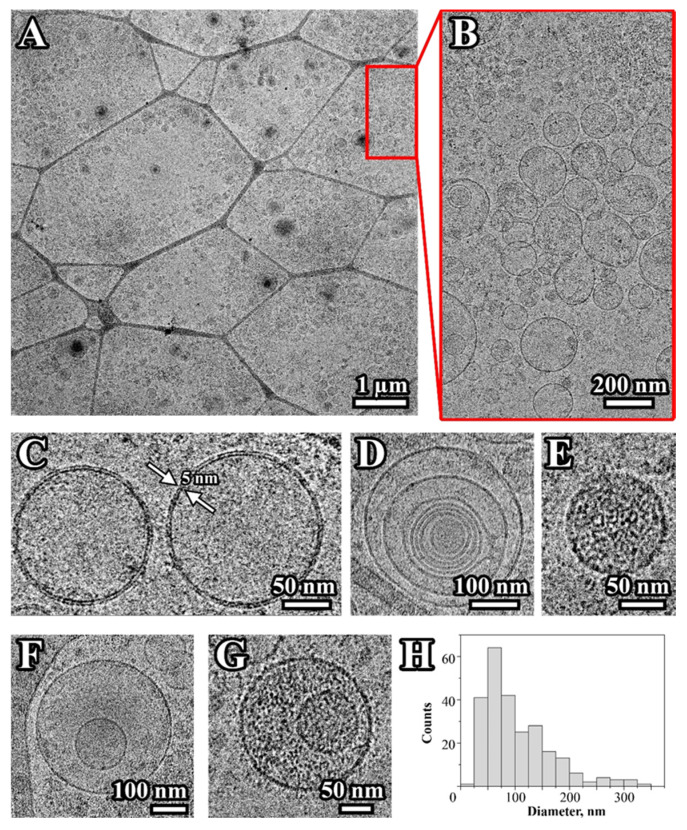
Cryo-EM images of vesicles isolated from CCM of glioma cells. Overview of the EV distribution on the cryo-EM grid (**A**,**B**); single (**C**), double (**F**), and multilayer vesicles (**D**); single and double vesicles containing electron-dense material (**E**,**G**). The size distribution of visualized vesicles (**H**). The arrows depict a lipid bilayer membrane of the vesicle. A total of 250 particles were analyzed.

**Figure 4 biomedicines-08-00216-f004:**
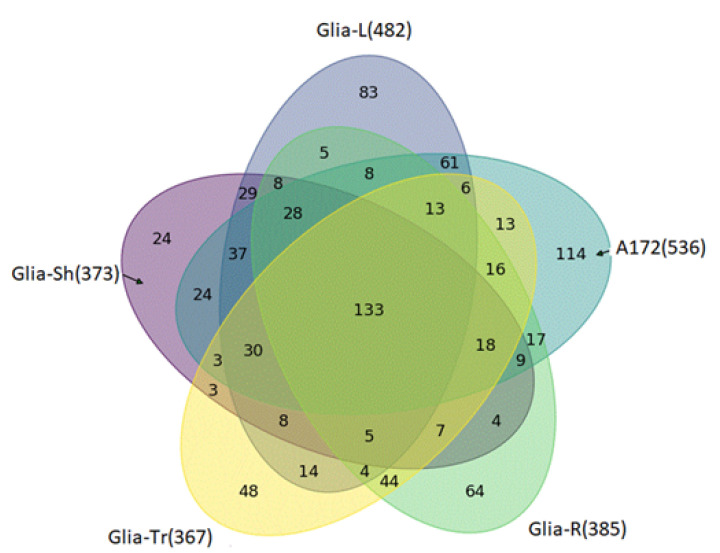
Venn diagram for proteins detected in all the 5 samples of exosomes.

**Figure 5 biomedicines-08-00216-f005:**
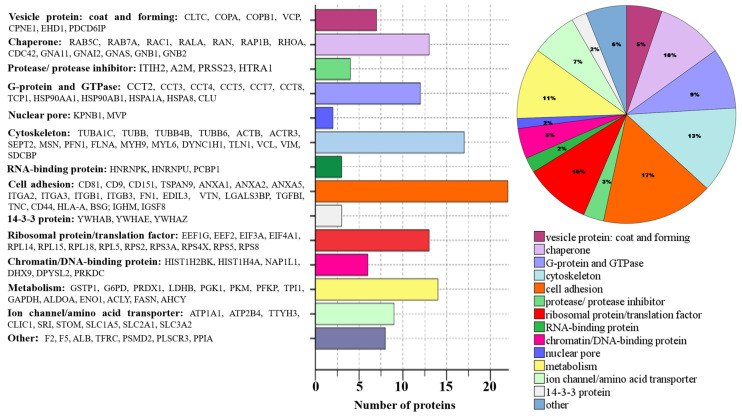
Molecular functions of 133 common proteins (the corresponding gene names are given) secreted by 5 glioma cell lines.

**Figure 6 biomedicines-08-00216-f006:**
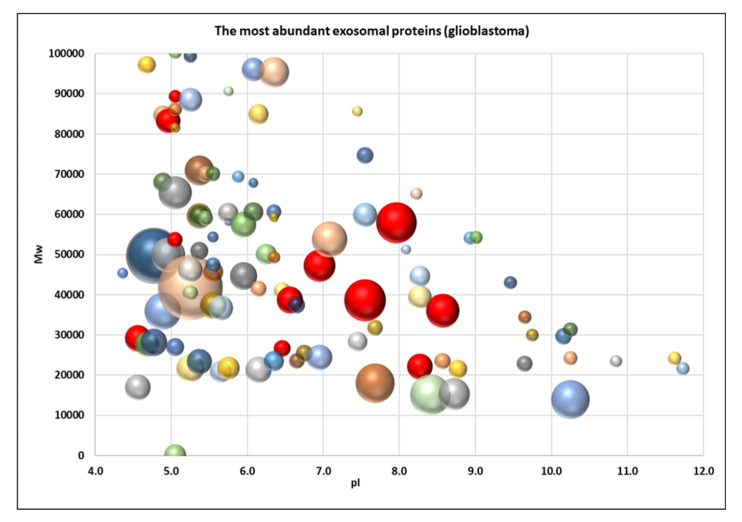
Virtual 2DE of the common exosomal proteins (Appendix A). Each ball represents a detected protein. The coordinates are pI/Mw, and the ball size is proportional to emPAI. The red balls represent proteins that are overexpressed in glioblastoma cells: KPYM, ANXA1, TPIS, VIME, ANXA2, TERA, ENOA, PRDX1, G3P, HS90B, 1433E.

**Figure 7 biomedicines-08-00216-f007:**
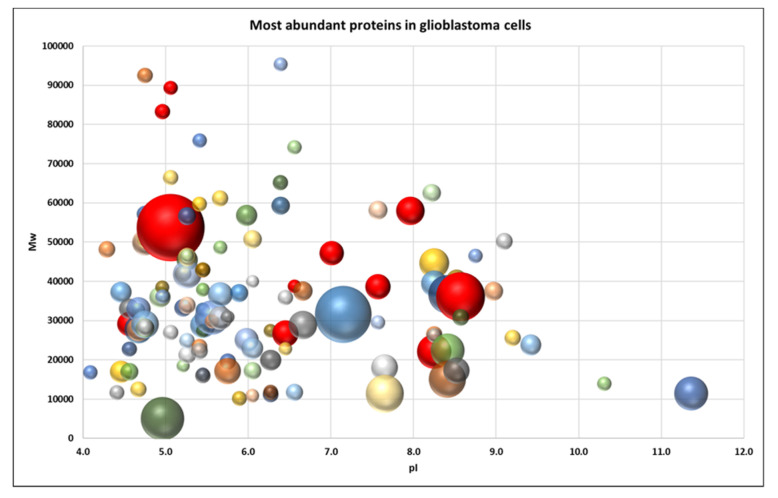
Virtual 2DE of the most abundant cellular proteins in glioblastoma (Appendix A). Each ball represents a detected protein. The coordinates are pI/Mw, and the ball size is proportional to emPAI. The red balls represent proteins that are overexpressed in glioblastoma cells: KPYM, ANXA1, TPIS, VIME, ANXA2, TERA, ENOA, PRDX1, G3P, HS90B, 1433E.

**Figure 8 biomedicines-08-00216-f008:**
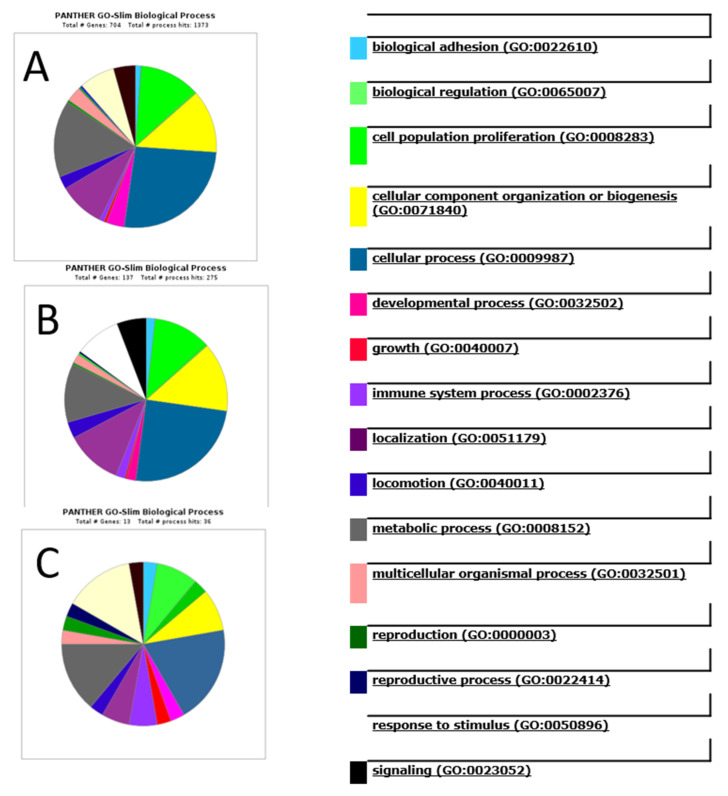
Analysis of the involvement of the detected proteins in biological processes using PANTHER software. (**A**) all the detected exosomes proteins; (**B**) the common exosomes proteins. (**C**) the potential glioblastoma biomarkers (ANXA1, ANXA2, ENOA, G3P, HS90B, KPYM, PRDX1, TPIS, TERA, VIME, 1433E, COF1, NPM) detected in exosome samples.

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
