# Peer review of "Proteome of Glioblastoma-Derived Exosomes as a Source of Biomarkers"

_biomedicines, 2020, doi:10.3390/biomedicines8070216_

Round 1
Reviewer 1 Report
In this study, Stanislav Naryzhny and co-authors, performed proteomics analysis of exosomes obtained from conditioned culture medium of glioblastoma cell lines (and glial cells??), to retrieve a protein list to be potentially used as glioblastoma biomarkers.
The EV isolation, characterization and proteomic analysis are well conducted and with proper control and the “protein list” the authors retrieved has relevant input to be further considered.
There are some major point to be addressed:
- The manuscript is quite confusing about the starting samples the authors used. In the paragraph 2.1 lines 64-65 they claim: “Five samples of vesicles isolated from conditioned culture medium (CCM) of glioma cell lines were analysed” but in the methods section they described the samples: “The work was carried out using A172 glioma cell line obtained from the collection of the Institute 189 of Cytology of the Russian Academy of Sciences (St. Petersburg, Russia). In addition, the study was 190 also performed using four primary glial cell cultures (Glia-Tr, Glia-L, Glia-R and Glia-Sh)”. So, it is totally unclear if in this study have been used five glioblastoma lines or one glioblastoma line and 4 Glial cells samples (I mean normal astrocytes). Throughout the manuscript it seems that the authors analysed 5 glioma samples, if so please explain the details in the methods sections. Moreover, if the 4 samples are glioma lines, did the authors performed the transcriptional subgrouping? It will be interesting to evaluate if there is any correlation between Glioblastoma specific transcriptional profile and EV- protein.
- Together with the manuscript the authors provided 3 excel file full of information. However, all these data are barely presented and discussed in the manuscript. In particular, protein list for each sample and the “common” protein list were used to generate Gene Ontology profiles. These profile should be reported and presented in the results section. Which are the most significant GO? Did they have any correlation with known biological processes in GBM? In the supplementary TABLE 2 authors showed HPA classes for each lines with a nice graph, these data are worthy to be reported in the results section and explained.
- Have the authors the chance to explore the possibility to find (and validate) some of potential identified biomarkers in GBM patients serum?
Minor comment:
The manuscript should be revise for the use of relative pronouns, introducing relative clauses.
Author Response
Many thanks to the reviewer for his/her comments. We hope that the manuscript revised according to these comments became better.
- Answer 1. We have been using 5 glioblastoma cell lines. Corrections in the Methods section were added to omit any confusions. Unfortunately, we didn’t perform the transcriptional subgrouping for these lines
- Answer 2. We added more GO data, graphs, explanation, and discussions in the text.
- Answer 3. We are working on potential identified biomarkers in GBM patients’ serum. But right now we cannot give any information about this issue.
- Answer 4. The manuscript was thoroughly checked and revised. English was corrected by a translation agency.
Reviewer 2 Report
In this study, the Authors have performed the proteomics analysis of exosomes obtained from conditioned culture medium of several glioblastoma cell lines and have identified the extracellular vesicles through technologically advanced analysis.
The analyzed topic is of great interest and the presented data are convincing, that if the authors could address the (minor) points below reported, the paper might be proposed for publication in Biomedicines.
Abstract section:
In the abstract there is a lot of general and not always relevant informations and little space is given to the presentation of the work, not allowing the reader to understand the aims and results of the work.
Results section:
It’s in my opinion that in the results section, the numerous comments and references to the literature should be moved to the part of the discussion, which is very poor.
The Authors report that the experiments have been performed on five different cell lines. However, in the Methods section, it is written that the study was also performed using four primary glial cell cultures. There is no information on methods for obtaining primary culture and their source (patients, informed consent, etc.)
The sentence (pag. 5, line 119): “Summarizing the data, it can definitely be stated that most vesicles isolated from CCM of glioma cells can be defined as exosomes, according to International Society for Extracellular Vesicles (ISEV) guidelines [18]” is strong statement and the reference appears not pertinent. The authors should provide more explanation.
Discussion sections
It’s in my opinion that the Authors should improve the discussion by taking up the various comments included in the results.
In the study there are no bibliographical references to scientific papers in which the effect of extracellular vesicles isolated from glioblastoma on cancer cells was studied.
Moreover, the authors should give further bibliographical details to demonstrate the validity of the methods used.
Author Response
Many thanks to the reviewer for his/her comments. We hope that the manuscript revised according to these comments became better.
- Answer 1. An abstract was completely rewritten.
- Answer 2. The discussion part was completely rewritten
- Answer 3. This part of the Methods section was corrected to omit confusions about cell lines.
- Answer 4. More details and explanation were added in the exosome section.
- Answer 5. The Results and Discussion part were completely rewritten. More references were added.
Round 2
Reviewer 1 Report
The authors fulfil the reviewer requests and provide explanation for the data they do not include in the manuscript.
Methods are now more detailed, and the results section is well organized and the data more explained and discussed.
The reviewer appreciates the figures added to the main manuscript.